# FlowX: Towards Explainable Graph Neural Networks via Message Flows

## Abstract

We investigate the explainability of graph neural networks (GNNs) as a step towards elucidating their working mechanisms. While most current methods focus on explaining graph nodes, edges, or features, we argue that, as the inherent functional mechanism of GNNs, message flows are more natural for performing explainability. To this end, we propose a novel method here, known as FlowX, to explain GNNs by identifying important message flows. To quantify the importance of flows, we propose to follow the philosophy of Shapley values from cooperative game theory. To tackle the complexity of computing all coalitions' marginal contributions, we propose an approximation scheme to compute Shapley-like values as initial assessments of further redistribution training. We then propose a learning algorithm to train flow scores and improve explainability. Experimental studies on both synthetic and real-world datasets demonstrate that our proposed FlowX leads to improved explainability of GNNs.

## 1 Introduction

With the advances of deep learning, graph neural networks (GNNs) are achieving promising performance on many graph tasks, including graph classification (Xu et al., 2019; Gao & Ji, 2019; Chen et al., 2020), node classification (Kipf & Welling, 2017; Veličković et al., 2018; Wu et al., 2019), and graph generation (Luo et al., 2021; You et al., 2018). Many research efforts have been made to develop advanced graph operations, such as graph message passing (Kipf & Welling, 2017; Veličković et al., 2018; Li et al., 2019), graph pooling (Yuan & Ji, 2020; Zhang et al., 2018; Ying et al., 2018), and 3D graph operations (Schütt et al., 2017; Klicpera et al., 2020). Deep graph models usually consist of many layers of these operations stacked on top of each other interspersed with nonlinear functions. The resulting deep models are usually deep and highly nonlinear and complex. While these complex systems allow for accurate modeling, their decision mechanisms are highly elusive and not human-intelligible. Given the increasing importance and demand for trustworthy and fair artificial intelligence, it is imperative to develop methods to open the black-box and explain these highly complex deep models. Driven by these needs, significant efforts have been made to investigate the explainability of deep models on images and texts. These methods are developed from different perspectives, including studying the gradients of models (Simonyan et al., 2013; Smilkov et al., 2017; Yang et al., 2019), mapping hidden features to input space (Zhou et al., 2016; Selvaraju et al., 2017), occluding different input features (Yuan et al., 2020a; Dabkowski & Gal, 2017; Chen et al., 2018), and studying the meaning of hidden layers (Yuan et al., 2019; Olah et al., 2018; Du et al., 2018), etc. In contrast, the explainability of deep graph models is still less explored. Since graph data contain limited locality information but have important structural information, it is usually not natural to directly extend image or text based methods to graphs. Recently, several techniques have been proposed to explain GNNs, such as XGNN (Yuan et al., 2020b), GNNExplainer (Ying et al., 2019), PGExplainer (Luo et al., 2020), and SubgraphX (Yuan et al., 2021), etc. These methods mainly focus on explaining graph nodes, edges, features, or subgraphs.

In this work, we observe and argue that message flows are the inherent functional mechanism of GNNs and thus are more natural and intuitive for studying the explainability of GNNs. To this end, we propose a message flow based explanation method, known as FlowX, to explain GNNs. FlowX attributes GNN predictions to message flows and studies the importance of different message flows. We first develop a systematic framework that lays the foundation on message flows for naturally explaining the message passing in GNNs. With our framework, the FlowX first quantifies

the importance of flows by following the phylosophy of Shapley values. Since message flows cannot be directly quantified to calculate marginal contributions, we propose an approximation scheme as the initial assessments of different flows. We then propose a learning-based algorithm taking advantages from the initial assessments to capture important flows in predictions. We conduct extensive experiments on both synthetic and real-world datasets. Experimental results show that our proposed FlowX outperforms existing methods significantly and consistently. Both quantitative and qualitative studies demonstrate that our proposed FlowX leads to improved explainability of GNNs.

## 2 RELATED WORK

### 2.1 GRAPH NEURAL NETWORKS

With the advances of deep learning, several graph neural network approaches have been proposed to solve graph tasks, including graph convolutional networks (GCNs) (Kipf & Welling, 2017), graph attention networks (GATs) (Veličković et al., 2018), and graph isomorphism networks (GINs) (Xu et al., 2019), etc. They generally follow a message-passing framework to learn graph node features. Specifically, the new features of a target node are learned by aggregating messages flows passed from its neighboring nodes. Without loss of generality, we consider the input graph as a directed graph with $n$ nodes and $m$ edges. The graph is denoted as $\mathcal{G} = (V, E)$, where $V = \{v_1, \ldots, v_n\}$ denotes nodes, and $E = \{e_{ij}\}$ represents edges in which $e_{ij}$ is the directed edge $v_i \rightarrow v_j$. Then it can be represented by a feature matrix $X \in \mathbb{R}^{d \times n}$ and an adjacency matrix $A \in \mathbb{R}^{n \times n}$. Each node $v_i$ is associated with a $d$-dimensional feature vector $x_i$ corresponding to the $i$-th column of $X$. The element $a_{ij}$ in $A$ represents the weight of $e_{ij}$, and $a_{ij} = 0$ indicates $e_{ij}$ does not exist. For the $t$-th layer in GNNs, the message aggregation procedures can be mathematically written as a two-step computation as

$$\text{Aggregate: } S^t = X^{t-1}\hat{A}^t, \tag{1}$$

$$\text{Combine: } X^t = M^t(S^t), \tag{2}$$

where $X^t \in \mathbb{R}^{d_t \times n}$ denotes the node feature matrix computed by the $t$-th GNN layer and $X^0 = X$. Here $M^t(\cdot)$ denotes the node feature transformation function at layer $t$ and $\hat{A}^t$ is the connectivity matrix at layer $t$. Note that we name the elements in $\hat{A}^t$ as layer edges and $\hat{a}_{jk}^t$ indicates the layer edge connecting node $j$ and $k$ in layer $t$. For example, in GCNs, the transformations are defined as $\sigma(W^t S^t)$ and $\hat{A}^t = D^{-\frac{1}{2}}(A + I)D^{-\frac{1}{2}}$ where $W^t \in \mathbb{R}^{d_t \times d_{t-1}}$ is a trainable weight matrix, $\sigma(\cdot)$ denotes the activation function, $I$ is an identity matrix to add self-loops to the adjacency matrix, and $D$ denotes the diagonal node degree matrix. We can stack $T$ GNN layers on top of each other to form a $T$-layer network, and the network function can be expressed as

$$f(\mathcal{G}) = g(M^T(M^{T-1}(\cdots M^1(X^0 \hat{A}^1)\cdots)\hat{A}^{T-1})\hat{A}^T).$$

When $f(\mathcal{G})$ is a graph classification model, $g(\cdot)$ generally consists of a readout function, such as global mean pooling, and a multi-layer perceptron (MLP) graph classifier. Meanwhile, when $f(\mathcal{G})$ is a node classification model, $g(\cdot)$ represents a MLP node classifier.

### 2.2 EXPLAINABILITY OF GRAPH NEURAL NETWORKS

A major limitation of GNNs is their lack of explainability. Thus, different methods have been proposed to explain the predictions of GNNs, such as GraphLime (Huang et al., 2020), GNNExplainer (Ying et al., 2019), PGExplainer (Luo et al., 2020), PGMExplainer (Vu & Thai, 2020), SubgraphX (Yuan et al., 2021), XGNN (Yuan et al., 2020b), and GraphSVX (Duval & Malliaros, 2021). These methods can be mainly grouped into four categories based on the views of their explanations. First, several techniques provide explanations by identifying important nodes in the input graph. For example, GradCAM (Pope et al., 2019) measures node importance by combining the hidden features and gradients; LRP (Baldassarre & Azizpour, 2019) and Excitation BP (Pope et al., 2019) decompose the predictions into several terms and assign these terms to different nodes; PGM-Explainer (Vu & Thai, 2020) builds a probabilistic graphical model by randomly perturbing the node features and employs an interpretable Bayesian network to generate explanations. Second, several existing methods, such as GNNExplainer (Ying et al., 2019), PGExplainer (Luo et al., 2020), and GraphMask (Schlichtkrull

et al., 2021), explain GNNs by studying the importance of different graph edges. These methods follow a similar high-level idea that learns masks to identify important edges while maximizing the mutual information. Next, the recent study SubgraphX (Yuan et al., 2021) proposes to explain GNNs via subgraphs. It incorporates the Monte Carlo tree search algorithm to explore subgraphs and employs Shapley values to measure the importance. Finally, XGNN (Yuan et al., 2020b) focuses on model-level explanations, which can provide high-level insights and general understanding. It proposes to generate graph patterns that can maximize a certain model prediction. While those methods explain GNNs from different views, none of them can provide explanations in terms of message flows and we believe message flows are more natural for performing explainability.

To the best of our knowledge, GNN-LRP (Schnake et al., 2020) is the only algorithm that explains GNNs by relevant walks. The relevant walk is defined as a $T$-length ordered edge sequence that corresponds to a $T$-step directed path on the input graph. To study walk explainability, GNN-LRP considers the GNN prediction as a function and decomposes it using higher-order Taylor expansions to distribute prediction scores to relevant walks. Specifically, by using $T$-order Taylor expansion with a proper root, each term in the Taylor expansion corresponds to a relevant walk and is regarded as the importance score. While our proposed FlowX shares a similar explanation target, i.e., flow/walk, with GNN-LRP, our method is fundamentally different. The GNN-LRP is developed based on score decomposition while our method follows the phylosopy of Shapley values from cooperative game theory as initial assessments and proposes a learning-based algorithm for the score generation. In addition, GNN-LRP has several constraints on the activation function and bias term used in GNNs while our method can be applied to general GNN models. Furthermore, as the GNN-LRP follows the Gradient $\times$ Input scheme, it may not pass the model parameter randomization test and may not be sensitive to model parameters (Adebayo et al., 2018).

**Differences with Other Methods Using Shapley Values:** Shapley values are commonly used in explaining machine learning methods. In particular, a recent study proposes a surrogate method, known as GraphSVX (Duval & Malliaros, 2021), to explain GNNs with both node and feature masks. Another recent study proposes SubgraphX (Yuan et al., 2021), which employs a search algorithm to explore and identify subgraphs with high Shapley scores. While these methods use Shapley values, there are several fundamental differences. First, our proposed method focuses on explaining message flows, which are the most basic and natural units for explanations as GNNs are based on message passing schemes. Second, we only use Shapley-like values as initial approximations to facilitating further training. Experiments show that the learning step is very important. Furthermore, the fundamental difference with GraphSVX is reflected in the fact that, while our method is a perturbation-based method, GraphSVX is a surrogate method (Yuan et al., 2020c). Due to these differences, we show in experiments that our FlowX achieves more natural and improved performance as compared with other methods.

## 3 THE PROPOSED FLOWX

While existing methods mainly focus on explaining GNNs with graph nodes, edges, or subgraphs, we propose to study the explainability of GNNs from the view of message flows. We argue that message flows are the fundamental building blocks of GNNs and it is natural to study their contributions towards GNN predictions. With our message flow framework, we propose a novel method, known as FlowX, to investigate the importance of different message flows. Specifically, we follow the phylosopy of Shapley values (Kuhn & Tucker, 1953) from game theory and propose an marginal contribution approximation scheme for them. In addition, a learning-based algorithm is proposed to improve the explainability of message flows.

### 3.1 A MESSAGE FLOW VIEW OF GNNS

We consider a deep graph model with $T$ GNN layers. Each GNN layer aggregates 1-hop neighboring information to learn new node embeddings. Hence, for any node, the outgoing messages are transmitted within its $T$-hop neighbors. Then the outputs of GNNs can be regarded as a function of such transmitted $T$-step messages, which are named as message flows in this work. Formally, we introduce the concept of message flows and message carriers as follow:

**Definition 1: Message Carrier.** We use the connectivity matrix to represent the carriers for message flows. Given the connectivity matrix $\hat{A}^t$ at layer $t$, the layer edge $\hat{a}_{ij}^t$ represents the message carrier with which the message passes from node $v_i$ to $v_j$ at layer $t$.

Note that we use the superscript $t$ to distinguish the message carriers in different layers since their corresponding message flows are different. Then the set of all message carriers, i.e., all layer edges, is defined as $\boldsymbol{\mathcal{A}} = \{\cdots, \hat{a}_{uv}^1, \cdots, \hat{a}_{uv}^t, \cdots, \hat{a}_{uv}^T, \cdots\}$ and $|\boldsymbol{\mathcal{A}}| = |E| \times T$.

**Definition 2: Message Flow.** In a $T$-layer GNN model, we use $\mathcal{F}_{ijk\ldots\ell m}$ to denote the message flow that starts from node $v_i$ in the input layer, and sequentially passes the message to node $v_j, v_k, \ldots, v_\ell$ until to node $v_m$ in the final layer $T$. The corresponding message carriers can be represented as $\{\hat{a}_{ij}^1, \hat{a}_{jk}^2, \ldots, \hat{a}_{\ell m}^T\}$.

In a $T$-layer GNN model, all message flows start from the input layer and end with the final layer so that their lengths equal to $T$. For the ease of notations, we introduce the wildcard $*$ to represent any valid node sequence and $\boldsymbol{\mathcal{F}}$ to denote the message flow set. For example, we can use $\boldsymbol{\mathcal{F}}_{ij*}$, $\boldsymbol{\mathcal{F}}_{*\ell m}$, and $\boldsymbol{\mathcal{F}}_{ij*\ell m}$ to denote the message flow sets that share the same message carrier $\hat{a}_{ij}^1$, $\hat{a}_{\ell m}^T$, or both of them, respectively. In addition, we employ another wildcard ? to denote any single node and ?$\{t\}$ to represent any valid node sequence with $t$ nodes. For example, $\boldsymbol{\mathcal{F}}_{?\{3\}}$ means the set of valid 2-step message flow with 3 nodes. Note that the following property of message flow sets also holds:

$$\boldsymbol{\mathcal{F}}_{ij*\ell m} = \boldsymbol{\mathcal{F}}_{ij*} \cap \boldsymbol{\mathcal{F}}_{*\ell m}. \tag{3}$$

The final embeddings of node $v_m$ are determined by all incoming message flows to node $v_m$, which can be denoted as $\boldsymbol{\mathcal{F}}_{*m}$. Since the output of the GNN model is obtained based on the final node embeddings, then it is reasonable to treat the GNN output as the combinations of different message flows. Hence, it is natural to demystify GNN models by studying the importance of different message flows towards GNN predictions.

## 3.2 SAMPLING MARGINAL CONTRIBUTIONS AS INITIAL ASSESSMENTS

While explaining GNNs with message flows seems to be promising, it is still crucial to properly measure the importance of those message flows. Hence, in this work, our FlowX proposes to follow the philosophy of Shapley value (Kuhn & Tucker, 1953) to use the marginal contributions in different flow sets as the initial assessments of flow importance. It is noteworthy that our initial assessments are not computing real Shapley values. Shapley value is a solution concept in cooperative game theory and used to fairly assign the game gain to different players. When considering marginal contributions in GNN explanation tasks, we treat the message flows as different players and the GNN prediction score as the total game gain. Formally, given the trained GNN model $f(\cdot)$ and the input graph $\mathcal{G}$, we use $\mathcal{F}_*$ to denote the set of all valid flows, *i.e.*, all players in the game. Then given any flow $\mathcal{F}^k$, we mathematically define the contribution of it as

$$\phi(\mathcal{F}^k) = \sum_{P \subseteq \boldsymbol{\mathcal{F}}_* \backslash \{\mathcal{F}^k\}} W(|P|)(f\left(P \cup \{\mathcal{F}^k\}\right) - f(P)), \tag{4}$$

where $W(|P|)$ is a weight function assigned to each term in the summation; $P$ denotes the possible coalition group of players, and $\phi(\cdot)$ denotes the flow score. Here $f\left(P \cup \{\mathcal{F}^k\}\right) - f(P)$ is the marginal contribution of flow $\mathcal{F}^k$ for a particular coalition group, which can be computed by the prediction difference between combining $\mathcal{F}^k$ with the coalition group $P$ and only using $P$. Note that Eq. 4 is equivalent to the classic Shapley value when $W(|P|) = \frac{|P|!(|\boldsymbol{\mathcal{F}}_*|-|P|-1)!}{|\boldsymbol{\mathcal{F}}_*|!}$ where $|\cdot|$ is the set size. To compute the flow score $\phi(\mathcal{F}^k)$, we need to enumerate all possible coalition groups and considers different interactions among players. However, it is time-consuming to consider all possible coalition; thus, we sample several marginal contributions to approximate the final flow score. Note that it is not possible to compute importance score with Eq. (4) for message flows since we cannot remove individual message flows from the GNN model and the finest component we can directly remove is the layer edge. For example, when removing the layer edge $\hat{a}_{ij}^1$, the whole flow set $\boldsymbol{\mathcal{F}}_{ij*}$ is removed from the model. In addition, enumerating all possible coalition groups is time-consuming when the input graph is large-scale and the GNN model is deep. Hence, in our FlowX, we propose an approximation scheme to compute Eq. 4 based on Monte Carlo (MC) sampling (Štrumbelj & Kononenko, 2014).

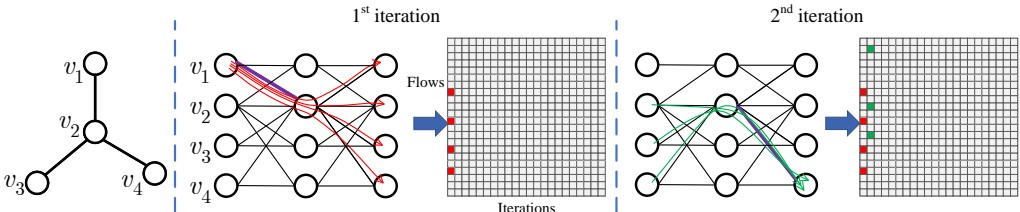

Figure 1: An illustration of our initial assessments via sampling marginal contributions. For each MC sampling step, we iteratively remove one layer edge until all layer edges are removed. In this example, the removed layer edges are shown in bold and purple lines while the corresponding message flows are shown in arrow lines. For example, in the first iteration, we remove the layer edge between $v_1$ and $v_2$ from the first GNN layer and compute the marginal contribution. Then three message flows are removed and the contribution scores are averaged and assigned to these three message flows.

---

**Algorithm 1** INITIAL APPROXIMATIONS OF FLOW IMPORTANCE SCORES.

1: Given a trained GNN model $f(\cdot)$ and an input graph, the set of all layer edges is represented as $\mathcal{A}$. For each message flow $\mathcal{F}^k$, two $|\mathcal{A}|$-dimensional vectors $S(\mathcal{F}^k)$ and $C(\mathcal{F}^k)$ denote its importance scores and removing index counts respectively. In addition, $\widehat{\mathcal{A}}$ denotes the set of removed layer edges and $\widehat{\mathcal{F}}$ is the set of removed message flows.

2: Initialize $S(\cdot)$ and $C(\cdot)$ as zeros for all message flows.

3: **for** step $i$ from 1 to Monte Carlo sampling step $M$ **do**

4:     Initialize the removed sets as empty that $\widehat{\mathcal{A}} = \emptyset$ and $\widehat{\mathcal{F}} = \emptyset$.

5:     Randomly shuffle and permute the layer edge set $\mathcal{A}$, denoted as $\mathcal{A}_{\pi(i)}$.

6:     **for** $j$ from 1 to $|\mathcal{A}|$ **do**

7:         Select the $j$-th layer edge in $\mathcal{A}_{\pi(i)}$, denoted as $\hat{a}_{\ell m}^t$.

8:         Block the layer edge $\hat{a}_{\ell m}^t$ in GNN model $f(\cdot)$, then the removed flows are $\widehat{\mathcal{F}}^j = \mathcal{F}_{?\{t-1\}\ell m?\{T-t\}} \setminus (\widehat{\mathcal{F}} \cap \mathcal{F}_{?\{t-1\}\ell m?\{T-t\}})$.

9:         Compute the prediction difference that $s_j = f(\mathcal{A} \setminus \widehat{\mathcal{A}}) - f(\mathcal{A} \setminus (\widehat{\mathcal{A}} \cup \hat{a}_{\ell m}^t))$.

10:        Update $\widehat{\mathcal{A}} = \widehat{\mathcal{A}} \cup \hat{a}_{\ell m}^t$ and $\widehat{\mathcal{F}} = \widehat{\mathcal{F}} \cup \mathcal{F}_{?\{t-1\}\ell m?\{T-t\}}$.

11:        Compute averaged score that $\bar{s}_j = s_j / |\widehat{\mathcal{F}}^j|$.

12:        **for** each flow $\mathcal{F}^k$ in $\widehat{\mathcal{F}}^j$ **do**

13:           Update $S(\mathcal{F}^k)[j] = S(\mathcal{F}^k)[j] + \bar{s}_j$.

14:           Update $C(\mathcal{F}^k) = C(\mathcal{F}^k) + 1$

15:        **end for**

16:     **end for**

17: **end for**

18: For each message flow $\mathcal{F}^k$, compute our marginal contribution vector $S(\mathcal{F}^k) = S(\mathcal{F}^k)/C(\mathcal{F}^k)$

---

Formally, let $M$ denote the total number of MC sampling steps. For the $i$-th sampling step, we use $\pi(i)$ to represent a random permutation of $|\mathcal{A}| = |E| \times T$ elements. Then the set of all layer edge $\mathcal{A}$ is permuted based on $\pi(i)$, denoted as $\mathcal{A}_{\pi(i)}$ that $\mathcal{A}_{\pi(i)}[j] = \mathcal{A}[\pi(i)[j]]$. In each step $i$, we iteratively remove one layer edge from $\mathcal{A}_{\pi(i)}$ following the order, and compute the marginal contribution. An illustration of our proposed sampling algorithm is shown in Figure 1. We use $\widehat{\mathcal{A}}$ to denote the set of removed layer edges and $\widehat{\mathcal{F}}$ to denote the set of removed message flows, which are initialized as $\widehat{\mathcal{A}} = \emptyset$ and $\widehat{\mathcal{F}} = \emptyset$ in the beginning of each MC sampling step. For the iteration $j$ in sampling step $i$, the $j$-th element of $\mathcal{A}_{\pi(i)}$ is removed and we assume the removed layer edge is $\hat{a}_{\ell m}^t$. Then the computation operations can be mathematically written as

$$s_j = f(\mathcal{A} \setminus \widehat{\mathcal{A}}) - f(\mathcal{A} \setminus (\widehat{\mathcal{A}} \cup \hat{a}_{\ell m}^t)), \tag{5}$$

$$\widehat{\mathcal{A}} = \widehat{\mathcal{A}} \cup \{\hat{a}_{\ell m}^t\}, \tag{6}$$

$$\widehat{\mathcal{F}}^j = \mathcal{F}_{?\{t-1\}\ell m?\{T-t\}} \setminus (\widehat{\mathcal{F}} \cap \mathcal{F}_{?\{t-1\}\ell m?\{T-t\}}), \tag{7}$$

$$\widehat{\mathcal{F}} = \widehat{\mathcal{F}} \cup \mathcal{F}_{?\{t-1\}\ell m?\{T-t\}}, \tag{8}$$

where $\widehat{\mathcal{F}}^j$ denotes the removed message flows in iteration $j$ by removing $\hat{a}_{\ell m}^t$. Note that it is not equivalent to $\mathcal{F}_{?\{t-1\}\ell m?\{T-t\}}$ since the flows in $\mathcal{F}_{?\{t-1\}\ell m?\{T-t\}}$ may be already removed in the previous iterations. Then the score is averaged that $\bar{s}_j = s_j / |\widehat{\mathcal{F}}^j|$ and we assign $\bar{s}_j$ to each flow in $\widehat{\mathcal{F}}^j$. By repeating such operations until all layer edges are removed, we can obtain $|\mathcal{A}|$ marginal scores and assign them to the corresponding flows. Note that the order information in $\mathcal{A}_{\pi(i)}$ is important since in the earlier iterations, the removed flows $\widehat{\mathcal{F}}^j$ are interacting with a larger coalition group $\mathcal{A} \setminus (\widehat{\mathcal{A}} \cup \hat{a}_{\ell m}^t)$ and in the later iterations the coalition groups are small. Altogether, the steps of our marginal contribution sampling are shown in Algorithm 1.

After $M$ MC sampling steps, our method explores $M$ permutations of layer edge set and each flow is sampled to obtain $M$ marginal contributions. For each flow $\mathcal{F}^k$, we use a $|\mathcal{A}|$-dimensional vector to store its marginal contributions, denoted as $S(\mathcal{F}^k)$, where $S(\mathcal{F}^k)[j]$ is the importance score obtained when $\mathcal{F}^k$ is removed in the iteration $j$. If the flow is removed mutiple times in the iteration $j$ for different MC sampling steps, the scores are averaged, which also means scores obtained from coalitions with same sizes are weighed equally. Then it raises the question that how to convert $S(\mathcal{F}^k)$ to the final importance score of $\mathcal{F}^k$. The final importance score will be a Shapley value approximation if we simply compute the summation of the elements in $S(\mathcal{F}^k)$. While Shapley values are promising, the approximations may not be accurate since $M$ is always set as $M << |\mathcal{A}|!$ for the sake of computation efficiency. Moreover, the importance of the early iterations when more layer edges exist in the model should be different to the importance of later iterations when only few layer edges exist in diverse situations. Hence, directly computing the summation of the scores in $S(\mathcal{F}^k)$ may not be suitable.

### 3.3 Learning Importance Scores

With our obtained score vectors $S(\mathcal{F}^k)$, we propose to regard these values as initial assessments of flow importance and learn an associated importance scores with designed redistribution trick. We target to learn weights for combining the elements in $S(\mathcal{F}^k)$ to obtain the final importance score for each flow, which means to learn $W(P)$ in Eq. 4.

Formally, given the input graph $\mathcal{G}$ and GNN model $f(\cdot)$, we obtain marginal contribution vector $S(\mathcal{F}^k)$ for each message flow. Then the only trainable weight vector $w$ is initialized to all equal to 0.5 with random noise, which shares the same dimension of $|S(\mathcal{F}^k)|$. Note that the weights $w$ are shared among all message flows in the same graph. Then $S(\mathcal{F}^k)$ can be combined via a dot product with $w$. Next, the flow importance is converted to layer edge importance by simply summing up the scores of all flows sharing a particular layer edge as the message carrier. Mathematically, it can be written as

$$s(\mathcal{F}^k) = S(\mathcal{F}^k) \cdot w \quad \forall \mathcal{F}^k \in \mathcal{F}_*, \tag{9}$$

$$s(\hat{a}_{uv}^t) = \sum_{\mathcal{F}^j \in \tilde{\mathcal{F}}} s(\mathcal{F}^j) \tag{10}$$

where $s(\cdot)$ denotes the importance score for layer edges and flows; $\tilde{\mathcal{F}} = \mathcal{F}_{?\{t-1\}uv?\{T-t\}}$. Then based on $s(\hat{a}_{uv}^t)$, we can obtain a mask indicating the importance of different layer edges, denoted as $\mathcal{M}$:

$$\mathcal{M} = g(\hat{s}), \tag{11}$$

where $\hat{s}$ denotes all layer edges' importance scores. $g(\cdot)$ is defined as an exponential redistribution operation that including an input normalization, an element-wise exponential scaling $g^e(x) = x^r$ ($r$ is a hyper-parameter), and an output normalization. We provide more intuitions in Appendix A.3. Note that the mask is normalized so that each element is in the range $[0, 1]$. By applying the mask to layer edges, important layer edges are restricted and the model prediction becomes

$$\hat{y} = f(\text{Combine}(\mathcal{G}, 1 - \mathcal{M})), \tag{12}$$

where $\hat{y}$ is the prediction vector and $\text{Combine}(\mathcal{G}, 1 - \mathcal{M})$ means layer edges are masked out based on the values of $\mathcal{M}$. Intuitively, if important layer edges are restricted, then the prediction should change significantly. Hence, $\hat{y}$ is encouraged to be different from the original prediction by learning proper

weights $w$. Specifically, we employ the negative cross-entropy loss that can be formally expressed as

$$\mathcal{L}(\hat{y}, y) = \sum_{c=1}^{m} \mathbf{1}\{y = c\} \log \widehat{y}_c, \tag{13}$$

where $y$ is a scalar representing the predicted class of the original graph $\mathcal{G}$, $m$ is the number of classes, $\mathbf{1}\{\cdot\}$ denotes the indicator function, and $\widehat{y}_c$ is the new predicted probability for class $c$. After training, the final importance scores for different flows can be obtained via Eq. (9).

## 4 EXPERIMENTAL STUDIES

### 4.1 DATASETS AND EXPERIMENTAL SETTINGS

**Datasets.** We employ seven different datasets to demonstrate the effectiveness of our proposed FlowX with both quantitative studies and qualitative visualization results. These datasets are BA-Shapes (Ying et al., 2019), BA-LRP (Schnake et al., 2020), ClinTox (Wu et al., 2018), Tox21 (Wu et al., 2018), BBBP (Wu et al., 2018), BACE (Wu et al., 2018), BA-INFE, and Graph-SST2 (Yuan et al., 2020c), which include both synthetic and real-world data. First, BA-Shapes is a node-classification synthetic dataset that is built by attaching house-like motifs to the base Barabási-Albert graph where the node labels are determined by their own identifies and localizations in motifs. Then, BA-LRP is a graph-classification synthetic dataset that includes Barabási-Albert graphs and the two classes are node-degree concentrated graph and evenly graph. Next, ClinTox, Tox21, BBBP, and BACE are real-world molecule datasets for graph classification. The chemical molecule graphs in these datasets are labeled according to their chemical properties, such as whether the molecule can penetrate a blood-brain barrier. Finally, Graph-SST2 is a natural language sentimental analysis dataset that converts text data to graphs. These graphs are labeled by their sentiment meanings. Note that the BA-INFE dataset is a synthetic dataset for Accuracy comparisons and the results are reported in the appendix. We also report the properties and statistics of these datasets in Appendix B Table 2.

**GNN Models.** In our experiments, we consider GCNs (Kipf & Welling, 2017) and GINs (Xu et al., 2019) as our graph models for all datasets. We adopt 2-layer GNNs for node classification and 3-layer GNNs for graph classification. The graph models are trained to achieve competitive performance and the details are reported in Table 2.

**Baselines.** With the trained graph models, we quantitatively and qualitatively compare our FlowX with eight baselines, including GradCAM (Pope et al., 2019), DeepLIFT (Shrikumar et al., 2017), GNNExplainer (Ying et al., 2019), PGExplainer (Luo et al., 2020), PGMExplainer (Vu & Thai, 2020), SubgraphX (Yuan et al., 2021), GNN-GI (Schnake et al., 2020), GNN-LRP (Schnake et al., 2020). Note that since these methods cannot be directly compared, we set the explanation target to graph edges for fair comparisons that the explanations are converted to edge importance scores if needed. More implementation details about explanation methods setting and GNN models can be found in the supplementary material. We used the datasets and implementations of the comparing algorithms in the DIG library (Liu et al., 2021). We will release our code after the anonymous review period.

### 4.2 QUANTITATIVE STUDIES

We first quantitatively compare different explanation methods. We follow the existing studies (Yuan et al., 2020c; Pope et al., 2019) and employ two metrics to evaluate the explanations: Fidelity and Sparsity. Good explanations should be faithful to the model and capture the discriminative features for the predictions. When such input features are removed, the original predictions should change significantly. The Fidelity score measures the change of predicted probabilities when removing important input features identified by different explanation methods. Higher Fidelity scores indicate the removed features are more important to the predictions and hence the explanations are more faithful to the model. In addition, the other desired property of explanations is sparsity. To encourage the explanations to be more human-intelligible, they should contain fewer but more important features. Hence, we also employ the Sparsity metric which measures the percentage of input features that are identified as important. Higher Sparsity scores indicate that fewer features are identified as important in the explanations. The formulations and details of these metrics are discussed in the appendix C.

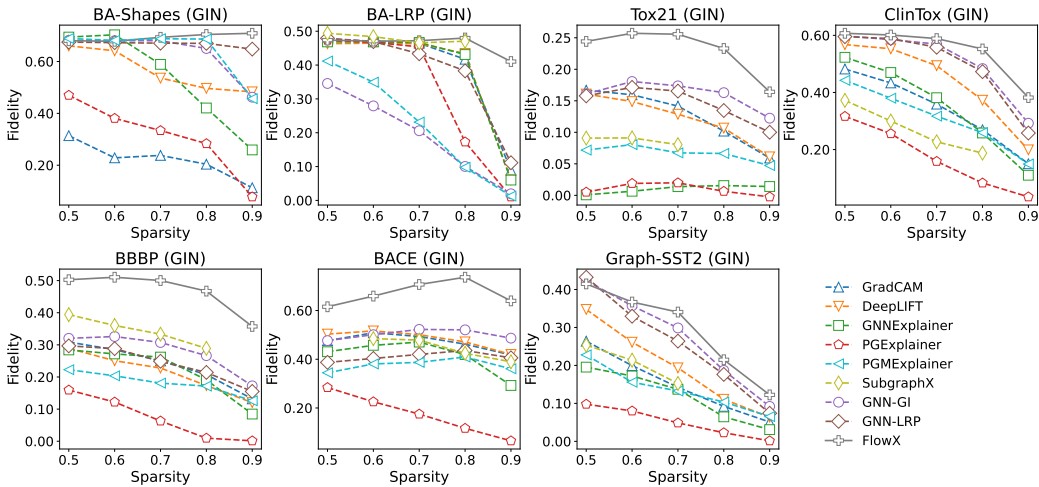

Figure 2: Comparison of Fidelity values by nine methods on seven datasets with GINs under different Sparsity levels.

Table 1: Results of ablation studies of our method on GCNs. FlowX$^\dagger$ denotes our FlowX without Shapley initial assessments and FlowX$^*$ denotes our FlowX without learning refinement.

|  | BA-Shapes | BA-LRP | ClinTox | Tox21 | BBBP | BACE | Graph-SST2 |
|---|---|---|---|---|---|---|---|
| FlowX$^\dagger$ | 0.40±0.03 | 0.17±0.10 | 0.34±0.08 | 0.17±0.05 | 0.39±0.17 | 0.32±0.06 | 0.22±0.09 |
| FlowX$^*$ | 0.41±0.00 | **0.52**±0.00 | 0.32±0.07 | 0.19±0.04 | 0.49±0.11 | 0.50±0.06 | 0.26±0.11 |
| FlowX | **0.42**±0.01 | 0.51±0.01 | **0.38**±0.06 | **0.22**±0.04 | **0.57**±0.11 | **0.51**±0.03 | **0.32**±0.13 |

It is noteworthy that these two metrics are highly correlated since the predictions tend to change more when more input features are removed. Then explanations with higher Sparsity scores tend to have lower Fidelity scores. Hence, we argue that the Fidelity scores need to be compared under a similar Sparsity level. Specifically, we control the Sparsity scores of explanations and compare the corresponding Fidelity scores. For each dataset, we randomly select 100 samples and conduct such quantitative evaluations. The results are reported in Figure 2 where we show the plots of Fidelity scores with respect to different Sparsity levels. Obviously, FlowX performs better on all of the real-world datasets that FlowX consistently achieves higher Fidelity across all Sparsity levels. Meanwhile, the results on synthetic datasets are competitive when the Sparsity is low and our FlowX still performs significantly better with high Sparsity levels. Note that our method is shown to achieve stable performance on different datasets and GNN models, which is promising for the applications and generalizations of our FlowX. Meanwhile, our proposed method, GNN-LRP, and GNN-GI perform better than the other methods, which indicates the superiority of the methods based on flows and walks. Note that GNN-LRP achieves good results on synthetic datasets but surprisingly, it is not better than GNN-GI on real-world datasets. In addition, PGExplainer only obtains competitive results on simple synthetic datasets but not on complex real-world datasets. Note that, for SubgraphX, its subgraph explanations are not directly comparable so we convert its subgraph explanations to edge explanations. It performs well on the BA-LRP and BBBP datasets but not as expected on the other datasets. We believe the reason may be that such converting destroys the continuousness and completeness of its subgraph explanations. Last but not least, we wish to mention that the comparisons using edge actually limit the performance of the flow-based methods due to explanation granularity changes. More results and time comparisons are also reported in the appendix.

## 4.3 ABLATION STUDIES

To demonstrate the effectiveness of our proposed learning-based algorithm in Section 3.3, we study the performance when only using the average of our Shapley initial assessments as the importance scores without further learning, denoted as FlowX$^*$. In addition, to show the Shapley initial assessments are

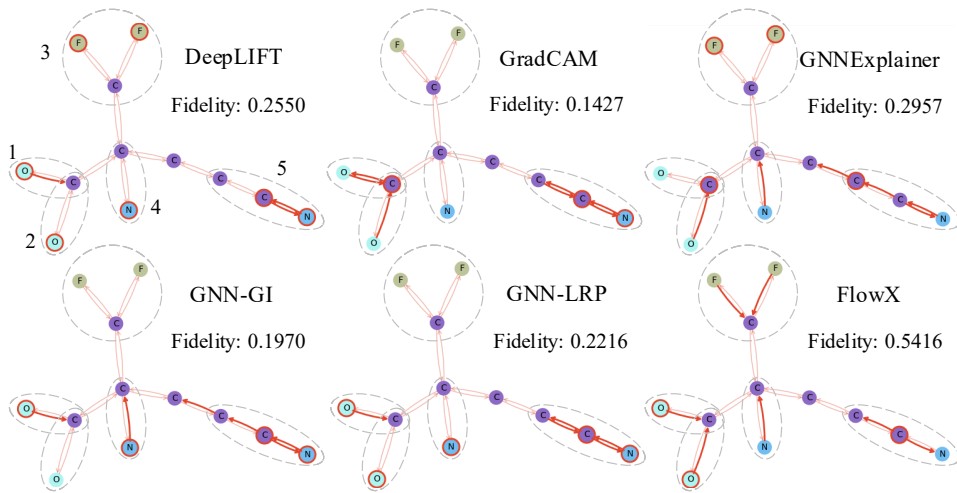

Figure 3: Sample explanation results of different methods. Note that explanations are shown as directed edges (highlighted red arrow lines) and self-loops (red circles around atoms). In addition, motifs are emphasized by dashed circles and numbered from 1 to 5.

necessary, we study the method only using the learning algorithm with randomly initialized scores for different flows, denoted as FlowX†. We compare our FlowX with these two baselines on GCN models, and the results are reported in Table 1. Obviously, for all datasets, the FlowX and FlowX* outperform FlowX† significantly, showing that the Shapley initial assessments are appropriate and necessary. In addition, for synthetic datasets, the FlowX and FlowX* show competitive performance, which indicates our initial assessments can obtain reasonable results for simple models and datasets. However, FlowX obtains significantly better performance on complex real-world datasets than FlowX*, showing the effectiveness of our proposed learning-based algorithm. More experimental details will be attached in the supplementary material.

## 4.4 VISUALIZATION OF EXPLANATION RESULTS

Finally, we report the visualization results of different explanation methods in Figure 3. Specifically, we show the explanations of a molecule graph from ClinTox dataset. The smiles string of this input graph is C(CC(C(F)F)(C(=O)[O-])[NH3+])C[NH3+] and the model to be explained is a 3-layer GCN model. In this input graph, motifs 1 and 2 form a Carboxylate Anion; motif 4 and 5 both contain an Ammoniumyl while motif 3 corresponds to a CF2 group. It is clear that our FlowX finds all motifs in the explanation and obtains the highest Fidelity score. It indicates that those motifs are indeed important for the model to make predictions. In addition, we can conclude that the model combines these motifs for the predictions since we generally observe that explanations with more motifs chosen tend to have higher Fidelity scores. Furthermore, we find that DeepLIFT identifies all motifs but focuses more on the self-loops of atoms. However, its Fidelity score is much lower than GNNExplainer who only identifies four motifs but focuses on the information transmissions between atoms. Hence, we believe the interactions among different atoms contribute more to the model predictions, which further indicates the Shapley value is a promising solution for studying the explainability of GNN models. More visualization results are reported in the supplementary material.

## 5 CONCLUSIONS

We study the explainability of deep graph models, which are generally treated as black-boxes. From the inherent functional mechanism of GNNs, we propose FlowX to explain GNNs by studying message flows. Our FlowX first computes Shapley values approximations as initial assessment and then incorporates a learning refinement. Extensive experiments demonstrate our FlowX achieves significantly improved explanations.

## REPRODUCIBILITY STATEMENT

Several efforts are made to ensure the reproducibility of our work. To demonstrate the details of proposed algorithm, we provide the rigorous description of our approach's first stage in Algorithm 1, the second stage learning process in Subsection 3.3 and the concrete setting in appendix B.4. For experiments, besides the detailed description of datasets and evaluation metrics in Section 4, we provide model configurations, training details, target mapping, and metrics details in appendix B and appendix C. Our implementations will be publicly available once the paper is published.

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

# A    ALGORITHM ANALYSIS & INTUITIONS

## A.1    MESSAGE FLOW INTUITION

Message flows can provide multi-hop dependence association, i.e., the reason of a given node why it is labeled as susceptible will be explained by a message flow beginning from an affected node to the given node in a virus infection dataset. Therefore, the explanation is that the message flow transmits the affection information from the affected node to the given node making it a susceptible one. The upper figures of Figure. 4 show that how a node's message can be transmitted from the node to which node across which path.

## A.2    COMPLEXITY ANALYSIS

**Message flows.**  Considering the total number of message flows $|\mathcal{F}|$, its loose upper bound is $|\mathcal{F}| = O(|E|^T)$. With the consideration of connectivity, given the largest outgoing degree of nodes in the graph $d_+$, the tighter upper bound of the number of message flows is $|\mathcal{F}| = O(|E|(d_+)^{T-1})$.

**Marginal contribution sampling.**  It is noticeable that the deep model's forward operations are the most time-consuming operations, denoted as $O(\mathcal{T}_f)$. Without any parallel consideration, the time complexity of marginal contribution sampling is $\mathcal{T}_{mcs} = O(M|\mathcal{A}|\mathcal{T}_f)$. When we consider parallel implementations, the most simple improvement is to use the GPUs acceleration to move line 9-15 out of the two loops and to make line 8 executed in $O(1)$, which leads to the time complexity $\mathcal{T}_{mcs} = O(M|\mathcal{A}| + \mathcal{T}_f)$. The space complexity should be considered because there is a big flow score matrix $\mathcal{S}_{cpu} = O(M|\mathcal{A}||\mathcal{F}|)$. As for the graphic memory requirement, this parallel implementation consumes $\mathcal{S}_{gpu} = O(M|\mathcal{A}|\mathcal{S}_f)$, where $O(\mathcal{S}_f) = O((|V| + |E|)dT + \mathcal{S}_p)$ is the deep model memory complexity; here, $|V| = n$ denotes the number of nodes; $\mathcal{S}_p$ is the model's parameter memory complexity. In our implementation, there is a trade-off that we only parallelly calculate the model outside the innermost loop, so that our time complexity is $\mathcal{T}_{mcs} = O(M(|\mathcal{A}| + \mathcal{T}_f))$ and the space complexity of GPUs is $\mathcal{S}_{gpu} = O(|\mathcal{A}|\mathcal{S}_f)$.

## A.3    TRAINING DILEMMA & EXPONENTIAL REDISTRIBUTION

We observe that the training is challenging as it often even leads to a worse result after training. The difficulty comes from that the training target is unusual. We not only require the important layer edges' scores to be trained to be their high-value local optimums with correct rankings but also require that all of the unimportant ones, not including those negative contribution elements, are trained to be low values because of the ranking requirement. The problem is that because the unimportant layer edges exert nearly no impact on the model results, their scores are trained to be the values that are largely random, which causes an unimportant layer edge's score may be larger than an important one because it is random (it may be arbitrary high or low). This is not what we expected. In brief, these random scores severely interfere with the ranking of scores of the important. Therefore, exponential redistribution is designed to solve this problem as shown in Eq. 11, where the gradients of the exponential scaling $g^e(x) = x^r$ decide that the high scores can be sensitively trained while the low scores are likely to be trapped in a low-value zone with gradients closed to zeros. Intuitively, considering the simplest 2-order exponential scaling $g^e(x) = x^2 = x \cdot x$, this function can be regarded as using $x$ as an attention map to mask itself so that high scores become easier to train while it is contrary to the low scores. It is noticeable that the use of exponential redistribution requires a relatively reasonable initialization, in order to give important elements a lower possibility to be initialized into the low-value zone, which leads to a longer training time. Hence, we initialize scores according to the marginal contribution results we obtain from the last stage so that unimportant factors are set into the low-value zone and will be likely to be trapped there having no interference to the high score ranking, while important layer edges are normally trained. Note that the final scores are not refined Shapley values, because the Shapley-like initialization only serves as heuristic values for the starting of exponential redistribution. Worth mentioning that the use of regularization terms like the size of masks and the information entropy does not work, and the use of Gumbel softmax for encouraging discreteness is useless either because both of them cannot solve the real problem.

Table 2: Statistics and properties of seven datasets. Note that "NC"denotes node classification, and "GC" denotes graph classification. # nodes (largest) denotes the number of nodes of the largest graph in the dataset for the split of explanations.

| Datasets | Task | # graph | # nodes (largest) | GCN Accuracy | GIN Accuracy |
|----------|------|---------|-------------------|--------------|--------------|
| BA-Shapes | Synthetic/NC | 1 | 700 | 90.29% | 89.57% |
| BA-LRP | Synthetic/GC | 20000 | 20 | 97.95% | 100% |
| BA-INFE | Synthetic/GC | 2000 | 39 | 99.00% | 99.50% |
| Clintox | Real/GC | 1478 | 136 | 93.96% | 93.96% |
| Tox21 | Real/GC | 7831 | 58 | 88.66% | 91.02% |
| BBBP | Real/GC | 2039 | 100 | 87.80% | 86.34% |
| BACE | Real/GC | 1513 | 73 | 78.29% | 80.26% |
| Graph-SST2 | Real/GC | 70042 | 36 | 90.84% | 90.91% |

## B  EXPERIMENTAL SETTINGS

### B.1  DEEP GRAPH MODELS

We first introduce the details of the GNN models we try to explain. For all GNN models, we employ two message passing layers for node classification (BA-Shapes (Ying et al., 2019)) and three for graph classification tasks (ClinTox, Tox21, BBBP, BACE (Wu et al., 2018), BA-LRP (Schnake et al., 2020), Graph-SST2 (Yuan et al., 2020c)). In addition, the final classifier consists of two fully-connected layers. For graph classification tasks, the average pooling is used to convert node embeddings to graph embeddings. In addition, the feature dimensions of message passing layers and fully-connected layers are set to 300. We apply the ReLU function as the activation function after each message passing layer and fully-connected layer. In addition, the dropout is applied between two fully-connected layers. Note that we consider both GCNs and GINs for all datasets. For the GCN layer, we employ the original normalized Laplacian matrix $\hat{A}^t = D^{-\frac{1}{2}}(A + I)D^{-\frac{1}{2}}$. For the GIN layer, two fully-connected layers with the ReLU function are employed as the multilayer perceptron (MLP).

### B.2  TRAINING SETTINGS

The models on different datasets use different learning rates and decay settings. Generally, we set the learning rate to $1 \times 10^{-3}$ and the learning rate decay equal to $0.5$ after 500 epochs. For datasets BA-LRP and Graph-SST2, the total number of epochs is 100 while we train models for 1000 epochs on the other datasets. All datasets are split into the training set (80%), validation set (10%), and testing set (10%). All experiments are conducted using one NVIDIA 2080Ti GPU.

### B.3  TARGETS OF EXPLANATIONS

Since different techniques focus on different explainable targets, such as nodes, edges, walks, flows, etc., these methods cannot be directly compared. For fair comparisons, we convert all explainable targets to graph edges. First, GNNExplainer and PGExplainer provide edge-level explanations so their results are directly used. Second, for flow-based methods and walk-based methods, including GNN-GI, GNN-LRP, and our FlowX, we convert the flow (walk) importance to edge importance by summing the total contribution of message flows (walks) that go through a particular edge. In addition, for node-based methods such as GradCAM (Pope et al., 2019) and DeepLIFT (Shrikumar et al., 2017), the node importance is mapped to edge importance that the contribution of an edge is the averaged contribution of its connected nodes. Specially, for subgraph-based method SubgraphX (Yuan et al., 2021), we pick the explainable subgraph out, then assign the edges in this subgraph instead of nodes as the explanation. Note that the absolute values of contribution are not important since the metrics focus on their relative rankings. The Figure. 4 shows how to map from message flows to edges.

### B.4  FLOWX SETTINGS

At the first stage, the MC sampling time $M$ is set as 30. At the second stage, we randomly initialize the weight vector of our learning refinement from the uniform distribution $[0, 0.1]$. During the training,

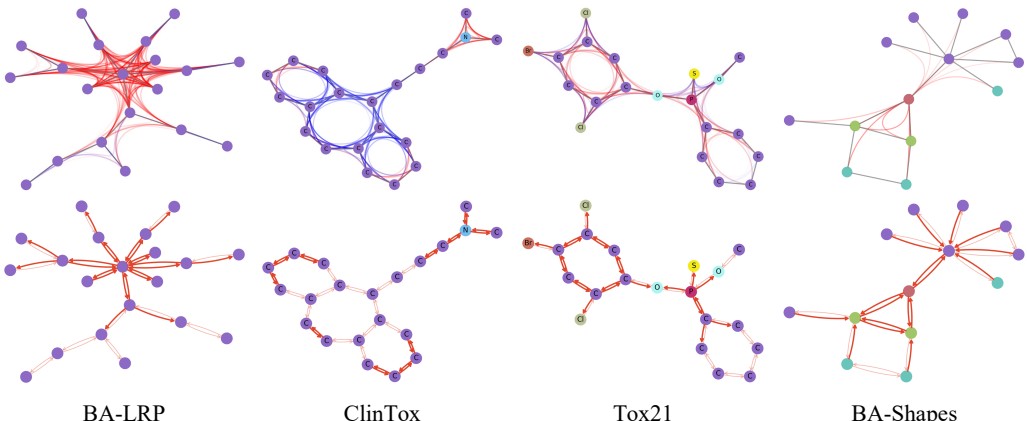

| BA-LRP | ClinTox | Tox21 | BA-Shapes |

Figure 4: An illustration showing how to convert message flow contribution to edge contribution. The top row shows the the flow view of explanations while bottom row shows the corresponding edge view. In the top row, we use red and blue flows to denote the positive and negative contribution respectively. In the bottom row, bold red arrow lines denote important edges.

we apply a learning rate equal to $0.3$ without learning rate decay and train the weight vector for $500$ iterations. Meanwhile, we adopt $r = 8$ for exponential redistribution.

**FlowX$^*$ & FlowX$^\dagger$.** FlowX$^*$ can be implemented by replacing the weight vector $w$ in section 3.3 with a constant vector $w'$ that $w' = 1/|w|$ where $|w|$ is the length of $w$. FlowX$^\dagger$ is implemented with a similar pipeline as GNNExplainer (Ying et al., 2019). Instead of training the weights to obtain the edge mask, we use a flow mask to indicate the flow importance scores. The flow mask can be considered as the weighted flow scores in our FlowX. In addition, we do not apply exponential scaling to FlowX$^\dagger$. The rest of FlowX$^\dagger$ is the same as the learning refinement part in our FlowX.

## C  EVALUATION METRICS

Mathematically, a $K$-class classification dataset with $N$ samples can be represented as $\{((X_i, \Lambda_i), y_i) \mid i = 1, 2, \ldots, N\}$, where $(X_i, \Lambda_i)$ is the input graph and $y_i$ is the ground-truth label of the $i$-th sample. The predicted class of a GNN model is $\hat{y}_i = \text{argmax} f(X_i, \Lambda_i)$. Here the output of $f(X_i, \Lambda_i)$ is a $K$-length vector, in which each element denote the probability of the corresponding class. Formally, we define the algorithm's explanation as a column vector $\dot{m}_i$ that represents a mask on explainable targets where the number of targets is $|\dot{m}_i|$. In this mask, the explanation method marks each selected target as a value $1$, otherwise a $0$. The source of the mask is a vector $\dot{c}_i$ with the same shape that stores the contribution score of each explainable target. Intuitively, we prefer to assign targets with $1$s in the $\dot{m}_i$ if the corresponding scores in $\dot{c}_i$ are relatively high. With these notations, we introduce the metrics employed for quantitative comparison.

**Fidelity.** Following the metrics mentioned in the graph explainability survey (Yuan et al., 2020c), we choose the metric Fidelity to measure the faithfulness of the explainability methods to the model. Mathematically, Fidelity is defined as:

$$\text{Fidelity} = \frac{1}{N} \sum_{i=1}^{N} (f(X_i, \Lambda_i)_{\hat{y}_i} - f^{\overline{\dot{m}}_i}(X_i, \Lambda_i)_{\hat{y}_i}), \tag{14}$$

where $f^{\dot{m}_i}(X_i, \Lambda_i)_{\hat{y}_i}$ denotes the output class $\hat{y}_i$'s probability when masking out the edges based on $\dot{m}_i$. Specifically, we keep the value unchanged for the explainable targets that are marked as $1$ in $\dot{m}$, while setting the other explainable targets as $0$. Note that $\overline{\dot{m}}_i = \mathbb{1} - \dot{m}$ is complementary mask where $\mathbb{1}$ is an all-one vector that has the same shape as $\dot{m}$. In the evaluation period, the higher Fidelity indicates the better performance of the explainability algorithm. Intuitively, Fidelity represents the predicted probability dropping after occluding the important features. Therefore, Fidelity can measure whether the chosen features are important to the predictions from the model's perspective.

Table 3: The averaged time cost of eight algorithms.

|  | GradCAM | DeepLIFT | GNNExplainer | PGExplainer | GNN-GI | GNN-LRP | SubgraphX | FlowX |
|---|---|---|---|---|---|---|---|---|
| Time (ms) | 14 | 15 | 353 | N/A | 704 | 5993 | 407784 | 4501 |

Table 4: Comparisons between FlowX and other methods in terms of average Fidelity over different Sparsity levels on GCNs. **Bold** and underline scores respectively denote the best and the second best results. In addition, because SubgraphX cannot control and reach all the Sparsity levels as we need, we cannot compare it with others in this table.

|  | BA-Shapes | BA-LRP | ClinTox | Tox21 | BBBP | BACE | Graph-SST2 |
|---|---|---|---|---|---|---|---|
| GradCAM | 0.39±0.09 | 0.50±0.03 | 0.23±0.07 | 0.10±0.02 | 0.30±0.12 | 0.29±0.06 | 0.18±0.08 |
| DeepLIFT | **0.42**±0.04 | 0.42±0.13 | 0.31±0.08 | 0.14±0.03 | 0.37±0.12 | 0.42±0.08 | 0.22±0.11 |
| GNNExplainer | 0.41±0.01 | 0.45±0.11 | 0.23±0.09 | 0.09±0.03 | 0.32±0.14 | 0.24±0.08 | 0.17±0.09 |
| PGExplainer | 0.38±0.07 | 0.43±0.12 | 0.20±0.10 | 0.02±0.00 | 0.14±0.07 | 0.10±0.05 | 0.07±0.04 |
| PGMExplainer | 0.38±0.06 | 0.35±0.16 | 0.25±0.06 | 0.08±0.01 | 0.25±0.06 | 0.17±0.01 | 0.16±0.06 |
| GNN-GI | 0.40±0.02 | 0.45±0.10 | 0.31±0.07 | 0.16±0.03 | 0.43±0.08 | 0.29±0.02 | 0.31±0.14 |
| GNN-LRP | 0.41±0.00 | **0.52**±0.01 | 0.32±0.08 | 0.14±0.03 | 0.35±0.10 | 0.25±0.05 | 0.21±0.11 |
| FlowX | 0.42±0.01 | 0.51±0.01 | **0.38**±0.06 | **0.22**±0.04 | **0.57**±0.11 | **0.51**±0.03 | **0.32**±0.13 |

**Sparsity.** In order to fairly compare different techniques, we argue that controlling the Sparsity (Pope et al., 2019; Yuan et al., 2020c) of explanations is necessary. Mathematically, Sparsity is defined as:

$$\text{Sparsity} = \frac{1}{N} \sum_{i=1}^{N} \left( 1 - \frac{\sum_{k=1}^{|\dot{m}_i|} \dot{m}_{ki}}{|\dot{m}_i|} \right). \tag{15}$$

A higher Sparsity score means fewer explainable targets are selected. Note that the selections of targets are determined by the ranking of the contribution score in $\dot{c}_i$. It is noteworthy that these two metrics are highly correlated since the predictions tend to change more when more input features are removed. Then explanations with higher Sparsity scores tend to have lower Fidelity scores. For fair comparisons, we compare different methods with similar sparsity levels.

## D EXPERIMENTAL RESULTS

In this section, we provide additional quantitative comparisons and visualization results.

### D.1 QUANTITATIVE RESULTS

We report the quantitative results of GCN and GIN models in Table 4 and Table 5. Obviously, our method obtains the best averaged Fidelity score for 5 out of 7 GCN models. Note that the performance is competitive for synthetic dataset BA-Shapes and BA-LRP and our method performs significantly better on the complex real-world datasets. In addition, we show the plots of Fidelity scores with respect to different Sparsity levels for all datasets and GCN models in Figure 5. Clearly, the performance of our method is stable and better.

**Accuracy comparisons:** To further demonstrate the effectiveness of our proposed method, we conduct experiments to compare the explanatory accuracy of three methods that usually evaluated in terms of accuracy. The results are reported in Table 6. These results are obtained from a synthetic graph classification dataset BA-INFE. Each graph includes a base BA-graph and four types of motifs. Two classes of motifs are attached (2 motifs denote the one property, 2 for another; let's denote them as class/property A and B). We first connect the same number (1 3) of each class's motifs to the base graph. Then we attach 1 to 3 extra motifs from one of the classes (for example, class A) to the base graph; thus, the number motifs in class A is more than class B, representing the graph tends to have the specific property (property A). The ground-truths of a graph are the elements in those motifs including edges (undirected) and self-loop connections. The number of ground-truth elements that are covered by explanation edges is denoted as the hit number. The accuracy is defined as (hit number) / (total number of ground-truth elements).

Table 5: Comparisons between FlowX and other methods in terms of average Fidelity over different Sparsity levels on GINs. **Bold** and underline scores respectively denote the best and the second best results. In addition, because SubgraphX cannot control and reach all the Sparsity levels as we need, we cannot compare it with others in this table.

| | BA-Shapes | BA-LRP | ClinTox | Tox21 | BBBP | BACE | Graph-SST2 |
|---|---|---|---|---|---|---|---|
| GradCAM | 0.22±0.06 | 0.38±0.15 | 0.34±0.12 | 0.13±0.04 | 0.24±0.06 | 0.47±0.03 | 0.15±0.08 |
| DeepLIFT | 0.56±0.07 | 0.38±0.16 | 0.44±0.14 | 0.12±0.04 | 0.21±0.06 | 0.48±0.03 | 0.20±0.10 |
| GNNExplainer | 0.53±0.17 | 0.38±0.16 | 0.35±0.15 | 0.01±0.01 | 0.22±0.07 | 0.41±0.06 | 0.12±0.06 |
| PGExplainer | 0.31±0.13 | 0.32±0.19 | 0.17±0.10 | 0.01±0.01 | 0.07±0.06 | 0.17±0.08 | 0.05±0.04 |
| PGMExplainer | 0.64±0.09 | 0.22±0.15 | 0.31±0.10 | 0.07±0.01 | 0.18±0.03 | 0.38±0.02 | 0.14±0.05 |
| GNN-GI | 0.63±0.09 | 0.19±0.12 | 0.51±0.11 | 0.16±0.02 | 0.28±0.06 | 0.50±0.02 | 0.27±0.12 |
| GNN-LRP | 0.67±0.01 | 0.38±0.14 | 0.49±0.13 | 0.15±0.03 | 0.24±0.05 | 0.41±0.02 | 0.26±0.12 |
| FlowX | **0.69**±0.01 | **0.46**±0.03 | **0.55**±0.08 | **0.23**±0.03 | **0.47**±0.06 | **0.67**±0.04 | **0.29**±0.11 |

Table 6: Comparisons between FlowX and three methods in terms of average Accuracy with 0.9 Sparsity on GCNs.

| | GNNExplainer | PGExplainer | PGMExplainer | FlowX |
|---|---|---|---|---|
| GCN | 0.2528 | 0.2199 | 0.3612 | 0.5266 |
| GIN | 0.2917 | 0.2143 | 0.3748 | 0.4265 |

We also show the time cost of different methods in Table 3. Specifically, we report the averaged computation time for 40 graphs in the ClinTox (Wu et al., 2018) dataset. It shows that our FlowX performs the best with a reasonable time cost.

## D.2 EXPLANATION VISUALIZATIONS

We provide additional examples to visually compare the explanations by six methods. The results are reported in Figure. 6, and 7. We can conclude that the message interactions between nodes are more important than self-loops. For example, in Figure. 6, all of DeepLIFT, GNNExplainer, and FlowX can identify the motifs; however, FlowX attributes more on the node interactions and obtain a higher Fidelity score.

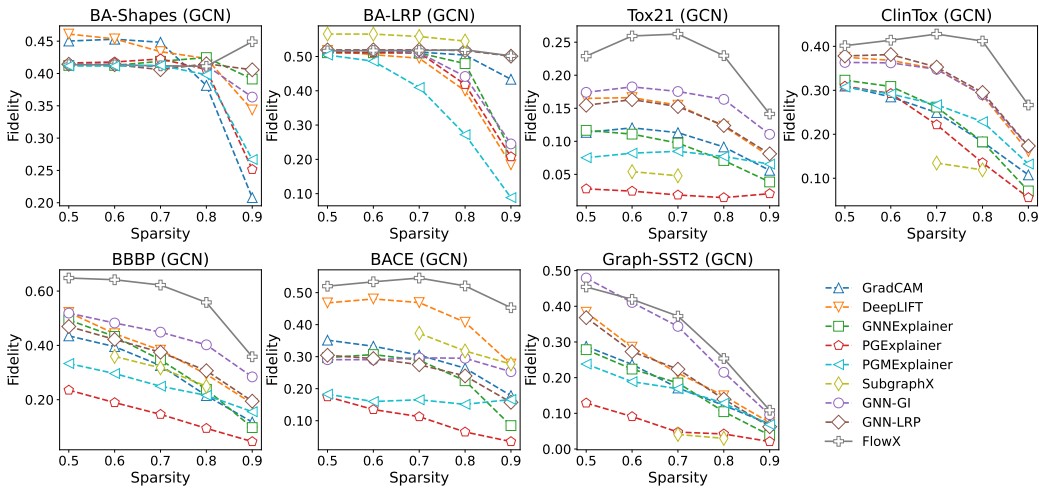

Figure 5: Comparison of Fidelity values by nine methods on seven datasets with GCNs under different Sparsity levels.

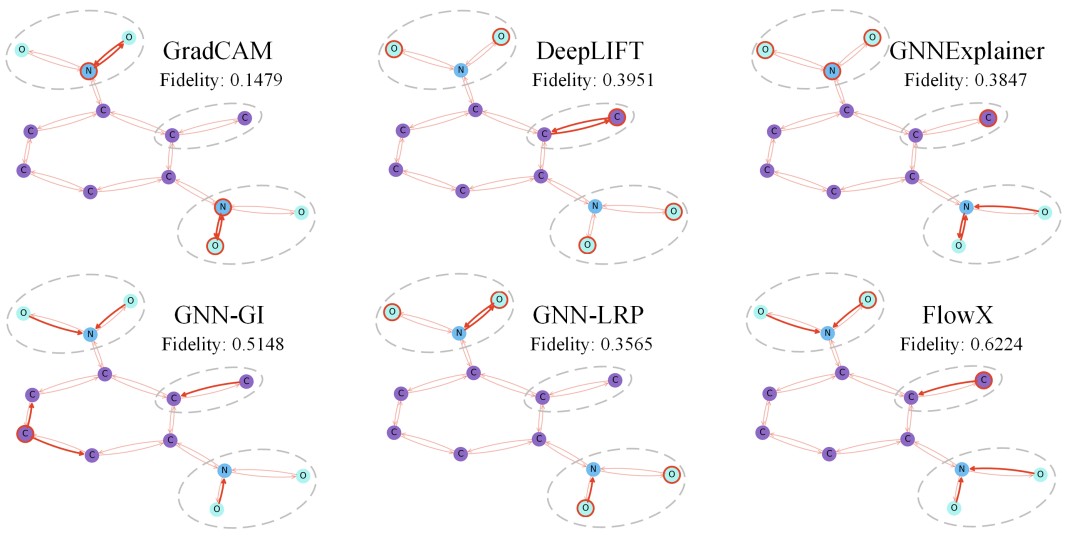

Figure 6: Sample explanation results of different methods on Tox21.

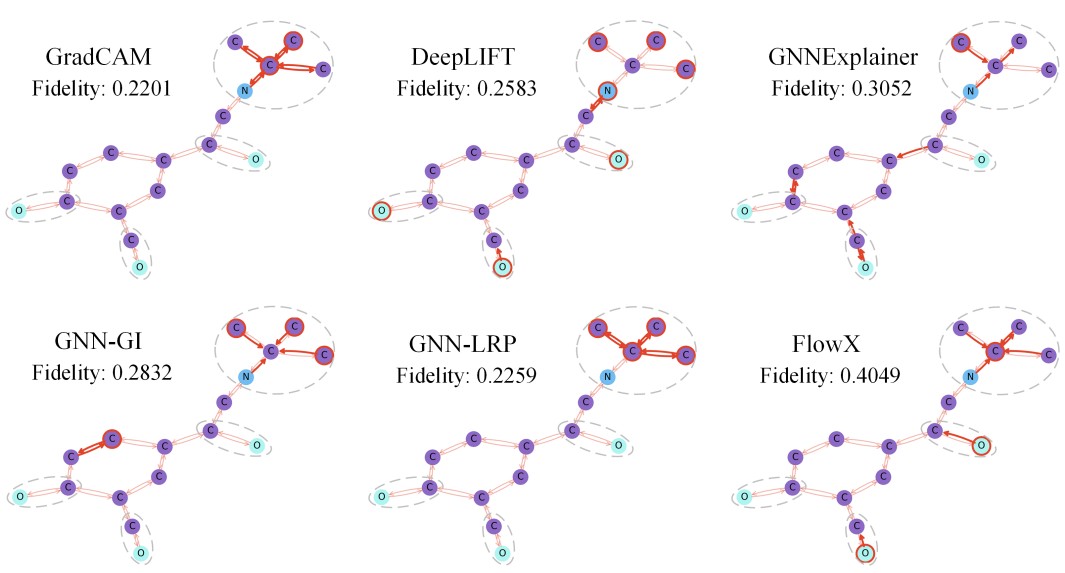

Figure 7: Sample explanation results of different methods on ClinTox.

