# OpenReview forum: "FlowX: Towards Explainable Graph Neural Networks via Message Flows"
_ICLR.cc/2022/Conference — ICLR 2022 Submitted_

### Official Review · Reviewer_bCqP · 2021-10-30

**Correctness:** 3
**Technical Novelty And Significance:** 3
**Empirical Novelty And Significance:** 2
**Recommendation:** 5
**Confidence:** 4

**Main Review:**

The paper presents a novel approach to address explainability in Graph Neural Networks but has several key points that remain unaddressed.

Strengths:
1. The proposed work is the first method that leverages message flow in GNNs to generate explanations for node- and graph-classification tasks.
2. Leveraging Shapley values for estimating the importance score of message flows is an interesting approach.
3. Extensive qualitative and quantitative experiments on both synthetic and real-world datasets.

Weaknesses:
1. The notations are inconsistent and confusing at times. For instance, $M$ denotes both the total number of Monte-Carlo sampling steps and the obtained mask from FlowX.
2. In the initial assessment using Shapley value approximations, the removal of one edge causes the removal of multiple message flows. The contribution scores are then uniformly averaged and assigned to the different message flows. Note that this can cause spurious distribution of scores causing higher importance to message flows that were not useful for a given node- or graph-classification task, especially in the final layers.
3. The notion of refining the Shapley importance scores is intuitive, but the use of optimization tricks like exponential scaling and encouraging discreteness can bias the refinement step to obtain edge masks that achieve better scores using the evaluation metrics in Sec. 4.
4. What does the weight vector $w$ represent in Eq. 9? Is it just some kind of mapper function that translates Shapley scores to edge mask scores?
5. The flow importances are converted to layer edge importance by simply summing up the scores of all flows sharing a particular layer edge as the message carrier. Would this cause the distribution of the layer edge importance scores to be skewed?
6. The authors did not use any baselines when comparing different explanation methods in Sec. 4 and why were the scores calculated using just 100 samples? Was it for computational complexity? If yes, then mentioning the error bar in the results would increase the significance of Sec. 4.
7. In Table 2, FlowX* performs on-par with FlowX on many datasets which questions the use of the refinement stage. An intuitive explanation for such behavior would be great.

Thank you for submitting a well-thought work!

**Summary Of The Paper:**

With explainability in Graph Neural Networks (GNN) still in a nascent stage, most graph explanation methods generate explanations in terms of nodes, node features, edges, or sub-graphs. GNN's are essentially message-passing networks, where every node has access to a local view of the graph created by the propagation of neural messages (embeddings) along edges in the node’s local neighborhood. However, none of the previously proposed GNN explainability methods consider message-flows as possible explanations. The paper proposes FlowX to identify important message flows by employing concepts of Shapley values, where an approximation scheme is used to estimate the Shapley values as initial assessments of flow importance. Finally, a learning algorithm is proposed to refine and map the message flow importance to edge mask explanations, where experimental studies on both synthetic and real-world datasets show the improvement of explanations using FlowX.

**Summary Of The Review:**

Check the main review.

---

> ### Author Response · Authors · 2021-11-17
> **Response to Reviewer bCqP (1)**
>
> Thank you for your comments. We now provide point-to-point responses to address your concerns.
>
> **Q: The notation problem:**
>
> **A:** Thank you for your comments. We have modified our paper to make it clear. Please check our updated paper.
>
> **Q: Concern that those unimportant flows are assigned higher scores than what they should have due to the average score distribution.**
>
> **A:** While we perform score averaging to assign contribution scores to different message flows, we believe it does not affect the results. First, even though unimportant flows can be assigned a high score in a particular iteration, important flows tend to have higher scores when considering multiple iterations since they can make their corresponding layer edges more important. While the absolute values of contribution scores may be affected, we care more about the score ranking of message flows. In brief, the probabilities for flows belonging to a high score layer edge are associated with their own importance, which will be revealed in the converged score expectation after many iterations. Furthermore, we wish to mention that since individual message flows cannot be directly removed, we believe our method is a reasonable way to approximate it.
>
>
> **Q: Encouraging discreteness can bias the training so that it will perform better using Fidelity as the metric. What is the intuition of the use of optimization tricks like exponential scaling?**
>
> **A:** We observe that the training is challenging as it often even leads to a worse result after training. The difficulty comes from that **the training target is unusual**. We not only require the important layer edges' scores to be trained to be their high-value local optimums with correct rankings but also require that all of the unimportant ones, not including those negative contribution elements, are trained to be low values because of the ranking requirement. The problem is that because the unimportant layer edges exert nearly no impact on the model results, their scores are trained to be the values that are largely random, which causes an unimportant layer edge's score may be larger than an important one because it is random (it may be arbitrary high or low). This is not what we expected. In brief, these random scores severely interfere with the ranking of scores of the important. Therefore, exponential redistribution is designed to solve this problem as shown in Eq. 11, where the gradients of the exponential scaling $g^e(x) = x^r$ decide that the high scores can be sensitively trained while the low scores are likely to be trapped in a low-value zone with gradients closed to zeros. Intuitively, considering the simplest 2-order exponential scaling $g^e(x) = x^2 = x\cdot x$, this function can be regarded as using $x$ as an attention map to mask itself so that high scores become easier to train while it is contrary to the low scores. It is noticeable that the use of exponential redistribution requires a relatively reasonable initialization, in order to give important elements a lower possibility to be initialized into the low-value zone, which leads to a longer training time. Hence, we initialize scores according to the marginal contribution results we obtain from the last stage so that unimportant factors are set into the low-value zone and will be likely to be trapped there having no interference to the high score ranking, while important layer edges are normally trained. Note that the final scores are not refined Shapley values, because the Shapley-like initialization only serves as heuristic values for the starting of exponential redistribution. Worth mentioning that the use of regularization terms like the size of masks and the information entropy does not work, and the use of Gumbel softmax for encouraging discreteness is useless either because both of them cannot solve the real problem.
>
> **Q. What does the weight vector $w$ represent in Eq. 9?**
>
> **A:** It is not just some kind of mapper function that translates Shapley scores to edge mask scores. It is associated with the $W(|P|)$ in Eq. 4 in the updated paper.
>
> **Q: Are the scores of layer edges derived by summing up the corresponding message flows fake?**
>
> **A:** As there is no precise way to convert contribution scores between flows and layer edges, we employ summation in our work as a good approximation. We argue that the absolute values are not important and we care about their relative ranking. From the ranking perspective,  we don’t regard them as skewed scores, while from the absolute distribution point of view, they can be considered as skewed.

---

> > ### Author Response · Authors · 2021-11-17
> > **Response to Reviewer bCqP (2)**
> >
> > **Q: The authors did not use any baselines when comparing different explanation methods in Sec. 4 and why were the scores calculated using just 100 samples? Was it for computational complexity? If yes, then mentioning the error bar in the results would increase the significance of Sec. 4.**
> >
> > **A:** We do compare our method with many baselines. There are 8 different baselines shown in Figure 2. Due to the space limitation, we put the complete quantity comparison tables in the appendix. Please refer to Tables 4 and 5. Therefore, 100 samples are enough considering the number of methods and datasets we used. For the time comparisons and complexity analysis, please refer to Table 3 and Appendix A. We have added the error bar in the results. Thank you for your advice.
> >
> > **Q: Why FlowX\* performs on-par with FlowX on many datasets?**
> >
> > **A:** FlowX* performs on-par with FlowX only on simple synthetic datasets with GCNs where most of the comparing methods also perform similarly (please refer to Table 4). However, for complex datasets, Flowx outperforms the rest significantly.

---

### Official Review · Reviewer_aAaM · 2021-11-01

**Correctness:** 3
**Technical Novelty And Significance:** 2
**Empirical Novelty And Significance:** Not applicable
**Recommendation:** 6
**Confidence:** 5

**Main Review:**

This paper tackles a significant problem (i.e., the explanation of Graph Neural Network) through a novel perspective: the importance of message flows. The paper is well-structured, with clear motivation and relation to existing baselines. It provides the reader with basic preliminary knowledge, describes the proposed method step by step, and evaluates its relevance. There are experiments with several datasets and baselines, as well as a small ablation study. So overall, I would say that this is an interesting work.

However, I am a bit skeptical about the authors’ approach, which involves computing the importance of message flows from edge importance in each layer of the GNN model, before converting it back to edge importance for each GNN layer—ultimately provided as an explanation. Indeed, since the marginal contribution of message flows cannot be directly computed via current GNN models, the paper calculates the marginal contribution of each edge for every GNN block. Then, it attributes this computed importance to all message flows containing this edge, without distinction; and repeats this for all possible edges and several Monte Carlo iterations. Finally, it re-defines GNN layer edge importance as the sum of each message flow containing those edges. As a result, in my opinion, considering message flows adds a great amount of computational complexity and storage requirement capacity while it can only bring limited additional information; since the finest granularity layer for GNN is edges in different layers.

Although this method seems very computationally expensive, the paper does not offer a proper complexity study, which appears essential. In addition to looping $M$ times on all edges of the graph in different GNN layers, it mentions storing matrices of size $|\mathcal{A}| = |E| \times T$ for each message flow, where the number of message flows is already difficult to track and grow exponentially with the number of edges. On the same note, the paper approximates all possible coalitions of players by sampling $M$ coalitions. There is no mention of how many Monte Carlo iterations $M$ are needed in practice (for the obtained results). Does $M$ relate to the size of the graph? Furthermore, at the end of Section 3.2, I believe there are $2^{\mathcal{|A|}}$ possible coalitions, not $|A|!$, since the order of element matters in Shapley values.

In Section 3.3, the paper proposes a learning-based algorithm to refine the initial assessments. In fact, to the best of my understanding, it learns to weight the marginal contribution of a player when added to a given coalition of players, depending on the size of the existing coalition. This extension, although nice and formulated in a learnable fashion, is not entirely novel as it was suggested by “A unified approach to interpreting model predictions” (Lundberg, 2017). This simplification aside, it would be interesting to see the distribution of the weight $w$ to see if a trend emerges—validating or not the intuition of SHAP.

**Interpretation.** The paper produces explanations for edges in different GNN layers. How can we leverage those, for instance, to spot which molecule is important for protein-function prediction? Or to spot if an Amazon product is a computer or not? The output explanation of the proposed method is not easy-to-understand for a human, especially as the number of GNN layers increases. Besides, it only targets graph structure and ignores features.

**Figure 1** seems not to be really helpful. It probably could have been more informative. Why are self-loops removed for simplicity? I do not believe it makes things more simple. Also, for node classification tasks, is the importance of a message flow from a node to itself really used?

**The notation is confusing.** For instance, $M$ refers to the number of MC samples as well as masks in 3.3. In Algorithm 1, it divides a vector by a vector (without saying it’s an element-wise division). In equation (9) it mentions element-wise operations, please mention dot product and do not confuse the reader with element-wise vector operation s(Hadamard product). The notation ‘For all’ in equation (10) is not rigorous, it should be in the sum.

**Regarding the evaluation.** The Accuracy metric is not used. I would have liked to see how FlowX compares to SOTA baselines on the evaluation conceived by GNNExplainer, and followed by most other baselines. Very few node classification datasets.

PGM-Explainer seems to be missing from Figure 1 - first graph. Also, why is no time associated with PGExplainer in the Supplementary Material?

How does FlowX's performance vary for different values of $M$? How is $M$ chosen in practice?



**Summary Of The Paper:**

FlowX is an explanation method for Graph Neural Networks. It derives importance measures for message flows, which are inherent function mechanisms of GNNs. These measures are initialized with a Monte Carlo approximation of the Shapley Values from Game Theory, applied to message flows. They are refined by learning a linear transformation, before being converted back to layer edge importance score ultimately provided as an explanation. Various experiments are performed to assess the relevance of the proposed explainer.

**Summary Of The Review:**

In a word, although this constitutes important work with a rather nice original intuition, the technical contribution is limited, the concept of Shapley Values was already leveraged by two GNN explainers (SubgraphX and GraphSVX -- which are not both used in the evaluation), and the evaluation is a bit light.

---

> ### Author Response · Authors · 2021-11-17
> **Response to Reviewer aAaM (1)**
>
> Thank you for your comments. We now provide point-to-point responses to address your concerns.
>
> **Q: Why do we need extra efforts to compute the importance of message flows with limited additional information.**
>
> **A:** First, Thanks to the low-layer GNN, it is possible to give message flow explanations. It is true that it has relatively high computation and storage requirements, but we still can do it with a reasonable time cost as shown in Table. 3. In addition, we can reduce the sampling steps without losing much performance. Second, it is interesting to show that the most important additional information that message flows provide is **multi-hop association**. For example, the cause of a given node why it is labeled as susceptible will be explained by a message flow beginning from an affected node to the given node in a virus infection dataset. Therefore, the explanation is that the message flow transmits the affection information from the affected node to the given node making it a susceptible one. With this multi-hop association, we will realize that many message flows go through the same layer edge but have different purposes and meanings.
>
> **Q: A requirement of time complexity analysis.**
>
> **A:** They are already added to appendix A.
>
> **Q: Does $M$ relate to the size of the graph? What is its value in practice?**
>
> **A:** It is not related to the graph in our setting. In practice, we find $M=30$ can be a good choice. It is added to appendix B.4.
>
> **Q: Why it is $M << |A|!$ instead of $M << 2^{|A|}$?**
>
> **A:** It is true that there are $2^{|A|}$ possible coalitions. But please be aware that the sampling process actually contains the accumulation of the same term leading to the coalition-size-related weights of terms in Eq. 4. Therefore, the total number of possible sampling is the permutation of $|A|$, which can be observed at line 5 in the Algorithm. 1.
>
> **Q: Will the distribution of the weight $w$ be trained to be similar to what it is in SHAP?**
>
> **A:** No. First, as we mentioned above, the sampling process actually has considered the Shapley weights, which means that FlowX* (section 4.3) is the method directly using Shapley weights. Therefore, the improvement from FlowX indicates that the trained weights are different from the original Shapley weights.
>
> **Q: Interpretation. How to use flow explanation reasonably?**
>
> **A:** Let’s consider a truck traffic map as an example to have some interpretations. The truck traffic map is corresponding to the graph, and we have the following mappings: roads (edges), crossroads (nodes), subzones (subgraphs), and flows(truck routes). By studying flow importance, we can know which truck routes are more important in this traffic map. In addition, we believe flow importance is very useful since by summing message flows, we can obtain different levels (edges/nodes/subgraphs) of explanations as it is the most fundamental building block in GNNs.  In addition, we only target graph structure but not features in this work because structures are more important for graphs.
>
> Here is another example.  The cause of a given node why it is labeled as susceptible will be explained by a message flow beginning from an affected node to the given node in a virus infection dataset. The explanation is that the message flow transmits the affection information from the affected node to the given node making it a susceptible one.
>
> It is true that our explanation is more aimed at explaining to computer scientists for further model debugging or understanding.

---

> > ### Author Response · Authors · 2021-11-17
> > **Response to Reviewer aAaM (2)**
> >
> > **Q: Figure 1: Why are self-loops removed for simplicity?**
> >
> > **A:** Thanks for the suggestion. We have modified it in the latest version paper.
> >
> > **Q: For node classification tasks, is the importance of a message flow from a node to itself really used?**
> >
> > **A:** Yes. For both graph classification tasks and node classification tasks, the self-loop message flows are always used.
> >
> > **Q: The notation problem.**
> >
> > **A:**  Thank you for your reminder. We have modified them in the newest version. Note that we also noticed that $C(\mathcal{F}^k)$ is denoted as a vector by mistake. We also correct it in our revised version.
> >
> >
> > **Q: Accuracy results.**
> >
> > **A:** We do provide the Accuracy results in Table. 6. Note that we believe that Fidelity comparison is more faithful to the model itself since Accuracy assumes that we have the correct explanation ground truths. However, even for synthetic datasets, the ground truths are approximated as we do not if the models can make predictions precisely in the expected way.
> >
> > **Q: why is PGM-Explainer for BA-Shapes in Figure 2 missing?**
> >
> > **A:** Thanks for the comment. We have added it in our revised version.
> >
> > **Q: Why is no time associated with PGExplainer?**
> >
> > **A:** PGExplainer time is missing because it has a different setting: it firstly trains all data then explains, instead of explaining each example individually. We believe it is
> > not comparable to other methods.
> >
> > **Q: Details about $M$.**
> >
> > **A:** FlowX’s performance becomes stable when $M=30$ in practice. We believe such a number is acceptable considering our superior performance.

---

### Official Review · Reviewer_ctC1 · 2021-11-01

**Correctness:** 4
**Technical Novelty And Significance:** 3
**Empirical Novelty And Significance:** 3
**Recommendation:** 6
**Confidence:** 5

**Main Review:**

Pros:
1). The idea is novel and interesting. Basically, to use message flows as the explanation is reasonable for GNNs.

2). The writing is clear and easy to follow.

3). The validation is also supportive for proof of the validity of the proposed framework.

Cons:
1). The initial generation of candidate sets of flows and the permutation algorithm seems to be time-consuming. The authors are required to analyze the time complexity of the proposed algorithm. Although, the authors list the computation time in the appendix, theoretical analysis of the time complexity is also useful. Moreover, in table 1, the author listed the number of graphs, however, to what extent the algorithm can handle large graphs. Thus, it is suggested to list the #nodes of the largest graph in the set.

2). The author adopts sparsity and fidelity as the metric for comparison. This is also used in SubgraphX. However, in PGExplainer and GNNExplainer, they use accuracy as the metric. It is suggested to also report the comparison of the accuracy of different methods.

3).  Typos and grammar errors:
maybe not sensitive to model==>maybe not be sensitive to model
may not suitable==>may not be suitable
Towards explanation of==>Towards an explanation of

**Summary Of The Paper:**

The paper proposed to identify important message flows as the explanation of the GNN models. To achieve acceptable fast calculation of Shapley values for identifying the message flows, the authors use n Monte Carlo (MC) sampling.

**Summary Of The Review:**

Basically, the idea is interesting and the solution is also sound. I would champion the acceptance. However, I have still some concerns as listed. Thus, I gave a weak acceptance.

---

> ### Author Response · Authors · 2021-11-17
> **Response to Reviewer ctC1**
>
> Thank you for your comments. We now provide point-to-point responses to address your concerns.
>
> **Q: A requirement of time complexity analysis.**
>
> **A:** Thank you for your suggestion. The time and space complexity analyses are added to Appendix A. Please refer to them for more details in the revised paper.
>
> **Q: A requirement of providing the nodes number of the largest graph.**
>
> **A:** The largest graphs of datasets for explanation splits are already added to Table. 2 in the Appendix of the revised paper. Thank you for your good idea.
>
> **Q: Some accuracy comparison is better to be preferred.**
>
> **A:** We already provided results in Accuracy with those methods usually using Accuracy as a metric. Please refer to Table. 6 in the Appendix.
>
> **Q: Typos and grammar errors.**
>
> **A:** We have corrected these typos in the latest version paper. Thank you for your reminder.

---

> > ### Comment · Reviewer_ctC1 · 2021-12-06
> > **Thanks for the feedback**
> >
> > Thanks for the feedback! Considering the concerns raised by other reviewers, I will keep my original rating.

---

### Official Review · Reviewer_ef9t · 2021-11-01

**Correctness:** 2
**Technical Novelty And Significance:** 2
**Empirical Novelty And Significance:** 1
**Recommendation:** 3
**Confidence:** 5

**Main Review:**

Strengths:

1. It is interesting that this paper uses important message flows as the explanation of GNN. Most existing methods use nodes, edges, or subgraphs to explain GNN.
2. The proposed method is tested on several datasets. Ablation study is employed to evaluate the importance of two stages. The experiments also show the time cost of different methods.
3. The paper is well structured and organized.

Weaknesses:

1. This paper lacks human intuition for using important message flows as an explanation of GNN. The practical significance of information flow as an explanation is not clear. Some practical examples may help readers to understand.
2. The contribution is somewhat incremental given existing explainers, for example, SubgraphX. Although a learning-based function is proposed to refine the importance score of flows, it is not sufficiently innovative.
3. In the experiments, some baselines are missing, such as Gem[1], GraphSVX.
4. In addition to using fidelity and sparsity as metrics, the accuracy of the explainer w.r.t. a ground truth explanation is also an important metric, which appears in GNNExplainer and other explanation-related papers.
5. Fidelity score performance on seven datasets with GCNs under different sparsity levels is inferior, which shows that the proposed method may not be effective.
6. In Table 4, comparisons between FlowX and other methods in terms of average fidelity over different sparsity levels are conducted. This comparison method is confusing and unconvincing to me. The average Fidelity score has no practical significance, and a high average fidelity score cannot indicate the effectiveness of the method.

[1] Wanyu Lin, Hao Lan, Baochun Li. Generative Causal Explanations for Graph Neural Networks. ICML 2021.

Minor points: The definition of s should be clear, s represents both marginal contribution and importance score. The definition of k in F_k is not clear.


**Summary Of The Paper:**

This paper explains GNNs by identifying important message flows. The authors use the concept of Shapley Value in Cooperative Game Theory and calculate Shapley Value as the initial assessments of flow importance score. A learning-based algorithm enables the refinement of important scores. Experiments are evaluated on both synthetic and real-world datasets.

**Summary Of The Review:**

Though this paper proposed a different way to explain GNNs, using message flows instead of nodes/ edges/ subgraphs, it lacks clear intuition of the novelty of using message flow. Relevant work is missing. Contributions are somewhat incremental compared with the existing Sharpley-value-based approaches for GNN explanation.

---

> ### Author Response · Authors · 2021-11-17
> **Response to Reviewer ef9t (1)**
>
> Thank you for your comments. We now provide point-to-point responses to address your concerns.
>
> **Q: Practical examples may help readers to understand the intuition of using message flows as explanations.**
>
> **A:** Message flows can provide multi-hop dependence association, i.e., the cause of a given node why it is labeled as susceptible will be explained by a message flow beginning from an affected node to the given node in a virus infection dataset. We have added more explanations in Appendix A.
>
> **Q: Because the method is still similar to SubgraphX, only refining the Shapley values of flows is not novel enough.**
>
> **A:** As we mentioned in Section 2, our method is fundamentally different from SubgraphX. Our method seeks to explain GNN by using message flows, which are the natural units in message passing neural nets, while SubgraphX explains GNNs using subgraphs. We believe message flows are more natural to GNNs. Shapley value is a common technique to evaluate the importance of units (just as Pearson correlation is commonly used to compute correlations), so it’s unreasonable to regard any methods with Shapley value as akin.
> More importantly, it’s a misunderstanding to regard our proposed method as Shapley-based methods. We did not actually calculate Shapley values and refine them. To eliminate the misunderstanding, we reform several equations and explanations in the latest version paper (changes are marked as red). Eq. 4 shows that the Shapley value is the special case for marginal contribution calculation. And the Shapley-like initialization is not simply refined to obtain the final score, instead, it is considered as the necessary requirement of exponential redistribution which can solve the training dilemma mentioned in the paper.
>
> Specifically, we observe that the training is challenging as it often even leads to a worse result after training. The difficulty comes from that **the training target is unusual**. We not only require the important layer edges' scores to be trained to be their high-value local optimums with correct rankings but also require that all of the unimportant ones, not including those negative contribution elements, are trained to be low values because of the ranking requirement. The problem is that because the unimportant layer edges exert nearly no impact on the model results, their scores are trained to be the values that are largely random, which causes an unimportant layer edge's score may be larger than an important one because it is random (it may be arbitrary high or low). This is not what we expected. In brief, these random scores severely interfere with the ranking of scores of the important. Therefore, exponential redistribution is designed to solve this problem as shown in Eq. 11, where the gradients of the exponential scaling $g^e(x) = x^r$ decide that the high scores can be sensitively trained while the low scores are likely to be trapped in a low-value zone with gradients closed to zeros. Intuitively, considering the simplest 2-order exponential scaling $g^e(x) = x^2 = x\cdot x$, this function can be regarded as using $x$ as an attention map to mask itself so that high scores become easier to train while it is contrary to the low scores. It is noticeable that the use of exponential redistribution requires a relatively reasonable initialization, in order to give important elements a lower possibility to be initialized into the low-value zone, which leads to a longer training time. Hence, we initialize scores according to the marginal contribution results we obtain from the last stage so that unimportant factors are set into the low-value zone and will be likely to be trapped there having no interference to the high score ranking, while important layer edges are normally trained. Note that the final scores are not refined Shapley values, because the Shapley-like initialization only serves as heuristic values for the starting of exponential redistribution. Worth mentioning that the use of regularization terms like the size of masks and the information entropy does not work, and the use of Gumbel softmax for encouraging discreteness is useless either because both of them cannot solve the real problem.

---

> > ### Author Response · Authors · 2021-11-17
> > **Response to Reviewer ef9t (2)**
> >
> > **Q: A requirement of adding Gem and GraphSVX as baselines.**
> >
> > **A:** After finishing the implementation of Gem, we believe that it is not comparable to our method. Here are some Fidelity average results on both GCN and GIN with 5 datasets:
> >
> > |  GCN   | BA-LRP         | ClinTox        | Tox21           | BBBP           | BACE           |
> > |:----|:---------------|:---------------|:----------------|:---------------|:---------------|
> > | Gem | 0.48$\pm$ 0.06 | 0.08$\pm$ 0.04 | -0.04$\pm$ 0.01 | 0.16$\pm$ 0.11 | 0.10$\pm$ 0.06 |
> >
> > |  GIN   | BA-LRP         | ClinTox        | Tox21           | BBBP           | BACE           |
> > |:----|:---------------|:---------------|:----------------|:---------------|:---------------|
> > | Gem | 0.37$\pm$ 0.17 | 0.06$\pm$ 0.05 | -0.03$\pm$ 0.01 | 0.14$\pm$ 0.08 | 0.20$\pm$ 0.06 |
> >
> > These results show that the Gem performs inferior to other methods in this setting, which is reasonable because after using Granger causal principle to obtain the ground truths, the GVAE model is actually a two-classification system so that the model cannot learn the edge scores (the ranking), instead, it can only justify whether an edge is important or not, depending on the ground truth generation stage. Hence, we choose to not compare with Gem in our paper for fairness evaluations.
> >
> > As we mentioned in the last sentence of Section 2,  GraphSVX has a totally different setting. It calculates node masks and node features, while we only care about the graph structure and study edge masks. It is not comparable between node/node feature masks between edge masks. While it is possible to convert node masks to edge masks in a straightforward way, such transformations will affect the performance of GraphSVX. Therefore, after careful considerations, we decide to not compare with it for fairness evaluations.
> >
> >
> > **Q: A requirement of Accuracy comparison, because it is important.**
> >
> > **A:** We already provide results in Accuracy with those methods usually using Accuracy as a metric. Please refer to Table. 6. But still, we believe that Fidelity comparison is more faithful to the model itself instead of a human-determined ground-truth since not all of the models are well-trained.
> >
> > **Q: The method may not be effective, because its performance with GCNs is not good.**
> >
> > **A:** This phenomenon only occurs in the two simple synthetic datasets with GCNs where most of the methods perform similarly, and our proposed Flowx has no more than 0.004 average Fidelity lagging behind, while in real-world datasets, we perform favorably against other methods with big gaps. Our method still shows its effectiveness and stability.
> >
> > **Q: Average Fidelity is confusing.**
> >
> > **A:** Average results are widely used when there are too many data points. It can be regarded as the area under the polyline in Figures. 2 and 5 with a scaling. Still, in order to obtain the most convincing results, please resort to Figures. 2 and 5 for original data points directly.
> >
> >
> > **Minor points:** Thanks for your suggestion about the symbol definitions. We have corrected them in the latest version paper.

---

### Comment · Reviewer_ef9t · 2021-12-06
**Rebuttal résponse**

Thanks for the feedback! Considering my concerns are not well addressed, I do not think this paper is qualified enough to be published in ICLR. I will keep my original rating.

---

### Decision · Program_Chairs · 2022-01-20

**Decision:**

Reject

**Comment:**

The paper proposes an explanation method based on message flows, and shows better performance than the state-of-the-art methods.
The authors addressed most of the reviewer's comments but the reviewers are not enthusiastic.  So I give my evaluation (some concerns are shared with reviewers and were not well addressed in the rebuttal)

Pros:
- State-of-the-art results on edge scoring.

Concerns:
- The main claim is not supported.  The authors say "we argue that message flows are more natural for performing explainability.  To this end, we propose..."  But I see no such argument after the proposed method is introduced.  Also, no advantage of the flow-based approach is shown.  The experiments only show edge scoring, which ignores the layer-wise edge scoring.  For this task, many existing methods are similar to the proposed approach in the sense that they measure how much information goes through subgraphs.  Although the proposed method shows good performance in edge scoring, this is not necessarily because the proposed method is flow-based, and cannot be evidence of the superiority of flow-based methods.  Fine details of the algorithm can contribute to the performance.  In the rebuttal, the authors mentioned a virus infection dataset as a situation where the flow-based method can do beyond what existing methods can do.  This kind of experiment should be shown in the paper to support the main claim.
- Difference between flows and walks is unclear.  The authors imply that this paper is the first paper based on the flows, and reviewers understood so.  The authors say a walk is "similar" to a flow but the difference is not explained.  (the authors only talk about the difference in how to compute the score in Schnake et al.)  Essential difference between walks and flows should be explained.
- The reason why the proposed method performs better than the existing methods is not analyzed.  The authors say they "believe" that this is because the proposed method is based on flows, but what readers want to see is evidence.
- Presentation should be improved.  Some formulations are unclear, e.g., I have no idea what F_?{t} means.  If this would be the best notation the authors think of, it should be explained with a figure.  Use another character if t is not the layer id.  Notation is not consistent, e.g., edges are denoted by e in Section 2.1 while they are denoted by a later.
- Marginal technical contributions.
- High complexity.  The proposed method seems not scalable even with the crude Monte Carlo approximation with a small number of samples.

With my concerns above and the reviewer's evaluations, I would not recommend acceptance.